# Computational Materials Design for Ceramic Nuclear Waste Forms Using Machine Learning, First-Principles Calculations, and Kinetics Rate Theory

**DOI:** 10.3390/ma16144985

**Published:** 2023-07-13

**Authors:** Jianwei Wang, Dipta B. Ghosh, Zelong Zhang

**Affiliations:** 1Department of Geology and Geophysics, Center for Computation and Technology, Louisiana State University, Baton Rouge, LA 70803, USA; 2Department of Geology and Geophysics, Louisiana State University, Baton Rouge, LA 70803, USA

**Keywords:** ceramic waste forms, apatite, hollandite, machine learning, first-principles calculations, rate theory, fission products

## Abstract

Ceramic waste forms are designed to immobilize radionuclides for permanent disposal in geological repositories. One of the principal criteria for the effective incorporation of waste elements is their compatibility with the host material. In terms of performance under environmental conditions, the resistance of the waste forms to degradation over long periods of time is a critical concern when they are exposed to natural environments. Due to their unique crystallographic features and behavior in nature environment as exemplified by their natural analogues, ceramic waste forms are capable of incorporating problematic nuclear waste elements while showing promising chemical durability in aqueous environments. Recent studies of apatite- and hollandite-structured waste forms demonstrated an approach that can predict the compositions of ceramic waste forms and their long-term dissolution rate by a combination of computational techniques including machine learning, first-principles thermodynamics calculations, and modeling using kinetic rate equations based on critical laboratory experiments. By integrating the predictions of elemental incorporation and degradation kinetics in a holistic framework, the approach could be promising for the design of advanced ceramic waste forms with optimized incorporation capacity and environmental degradation performance. Such an approach could provide a path for accelerated ceramic waste form development and performance prediction for problematic nuclear waste elements.

## 1. Introduction

The sustained development of nuclear energy requires the safe disposal of radionuclides produced from nuclear fission. Depending on nuclear fuel cycle options, which may involve several processes such as enrichment, off-gas capture, separation, and reprocessing, different waste streams with quite different compositions and levels of toxicity may result. Nuclear waste forms, including borosilicate glass and ceramics, are developed to immobilize these radioactive elements. However, due to factors such as the processing temperature of the glass, incorporation capacity, and long-term chemical durability in aqueous environments, borosilicate glass cannot efficiently immobilize certain radionuclides such as I-129, Cs-137, Cs-135, Tc-99, and Cl-36. If these radionuclides are released into the environment, they most likely form relatively large ionic species such as I^−^, Cs^+^, TcO_4_^−^, and Cl^−^ in most natural aqueous environmental conditions, due to their large stability field in Eh-pH phase space. Because of their low ionic potential (formal charge to radius ratio), they are less likely to form insoluble compounds through interactions with rocks and other dissolved species in the environment. Thus, they are mobile with a very limited amount of adsorption on the surfaces of rocks in the disposal environment. Furthermore, the surfaces of many silicate minerals, common in the disposal environment, are negatively charged because their pH values for the surface point of zero charge (PZC) are usually lower than the pH values of the groundwater expected to occur in the nearby field and geological formations [1,2,3]. For instance, for felsic (acidic, more silica content) igneous rocks (granite and rhyolite) consisting of minerals including quartz, albite, orthoclase, muscovite, and their weathered minerals such as kaolinite, their PZCs are from 1.5 to 4.8 [4]. For mafic rocks such as gabbro and basalt (basic, less silica content), the PZCs of the constituent minerals such as pyroxene and forsterite become higher, at 7.5 and 9.0, respectively [4,5]. However, these basic rocks often consist of a substantial amount of altered minerals with high surface area such as serpentine, montmorillonite, and chlorite (also common in sedimentary rock) with a PZC less than 5 [5]. As a result, these negatively charged surfaces cause negligible surface adsorption of the negatively charged aqueous ions, such as I^−^, Cl^−^, and TcO_4_^−^, on the silicate materials because the species and the surfaces are both negatively charged. As such, these ionic aqueous species are highly mobile in the environment. Therefore, these abovementioned elements are considered to be the most problematic radionuclides, which have the greatest potential of long-term adverse impact on the environment. It is therefore essential that appropriate waste forms are developed to efficiently incorporate these elements with sufficiently high loading and high chemical durability in the environment. 

Due to their high structural compatibility with certain nuclear waste elements, including long-lived fission products, actinides, and activation products, crystalline phases have been selected to develop ceramic waste forms that immobilize a group of radionuclides, such as synroc, or a single radionuclide, such as iodoapatite [6,7,8,9,10,11,12,13,14,15]. Unlike glass waste forms, ceramic waste forms have existing technical challenges and are not omnipotent in the immobilization of all waste components, but they are effective in targeting certain problematic radionuclides. For instance, the iodoapatite (Pb_5_(VO_4_)_3_I) waste form was developed to incorporate iodine-129 (~16 million years of half-life), which has very low solubility in nuclear glass, because apatite’s crystallographic channel site can accommodate large anions such as iodide [16,17,18,19,20,21,22]. Apatite-structured materials (materials with apatite structure but with a composition different from natural apatite) are also capable of incorporating cesium, strontium, chlorine, rare earth elements, and some actinides [21]. Similarly, sodalite-structured materials can incorporate iodide in the β-cage site of the crystal structure [23,24]. Hollandite and pollucite waste forms have been developed to incorporate Cs^+^ in its large channel site and 12-coordinated crystallographic sites, respectively [25,26,27,28,29,30,31,32,33,34]. In addition, other crystalline ceramic waste forms such as perovskite, pyrochlore, murataite, monazite, and crichtonite have been developed for the incorporation of various radionuclides, including Cs, Sr, rare earth, and actinides [11,35]. A great advantage of these selected crystalline phases is that each of them has a flexible composition and is capable of incorporating multiple elements by chemical substitution. These synthetic ceramic phases also have natural analogues and are chemically durable in the environment [11,35,36,37]. While their compositional flexibility is intriguing in pursuing correlation and coupling relationships between composition, structure, and properties, and provides a great opportunity to explore the compositional space to improve radionuclide immobilization, it also presents an enormous challenge for optimizing the composition of a given waste form with respect to waste element loading capacity and waste form environmental performance. This is because, as these phases have multiple substitution sites, the possible number of combinatorial substitutions at different sites and the number of materials to be tested become enormous when considering potential substituting elements from the periodic table. Therefore, approaches based on the trial-and-error method of testing the possible compositions require a substantial amount of work and decades of research and development. In this regard, methods such as machine learning and other computational methods that are capable of processing a large number of compositions for a given crystalline phase are essential in identifying and narrowing down the potential compositions that are worthy of further consideration.

In addition, ceramic waste forms need to be chemically durable in aqueous environments to ensure their long-term performance in a repository. It would be ideal if waste forms could be designed by concurrently considering both radionuclide incorporation and long-term chemical durability in aqueous environments. While it is not feasible to test the chemical durability of the waste forms under all conditions that may occur in a repository over a time period of hundreds of thousands of years, test protocols such as the Product Consistency Test [38], which was designed to evaluate and screen nuclear waste forms, may not be able to account for chemical durability under different conditions and over a long period of time, as degradation mechanisms may change with the degree of degradation and time [37,39]. Therefore, a systematic test of the chemical durability, aiming to understand the dissolution mechanism, is necessary for the reliable prediction of long-term performance under various environmental conditions. Based on an understanding of the dissolution mechanisms, modeling methods such as those based on thermodynamic kinetic theories that relate the dissolution rate to environmental conditions are necessary for the prediction of the chemical durability over time. Only experiments or computations that are critical to parameterizing the kinetic rate equations are needed. Once the kinetic rate equations are properly parameterized, they can be used to predict dissolution rates under various environmental conditions.

As described above, the rapid development of crystalline ceramic nuclear waste forms requires handling a large number of compositions for a given ceramic phase, and the waste element loading and dissolution rates of these waste forms in the compositional space need to be characterized. These needs call for an efficient computational approach, which is the focus of this review. However, due to the large scope of computational material design and nuclear waste development in the broad literature, a comprehensive review of computational material design for nuclear waste forms is not intended. Instead, the purpose of this review is rather narrow and only focuses on some of our most recent progress in the design of ceramic waste forms using machine learning, first-principles calculations, and the modeling of dissolution kinetics of crystalline ceramics using thermodynamic kinetic rate theory. Apatite-structured and (to a lesser extent) hollandite-structured materials will be used as examples and model systems to highlight this progress. Waste forms for specific waste, or waste streams such as long-lived and highly radio toxic actinides, are not emphasized, but they are very important to nuclear waste management. Together, these advances provide examples to demonstrate that these methods can be employed to accelerate the development of ceramic waste forms that specifically target some of the aforementioned problematic radionuclides. Note that there are significant developments in ceramic waste forms that will not be discussed in this review, but if interested, please refer to some of sources in the broader literature [6,7,8,9,10,11,12,13,35,36,37,39,40,41]. With advances in computational material design from the scientific community [42,43,44,45,46,47,48,49,50,51] and progress in the computation and mechanistic understanding of elemental incorporation and dissolution kinetics, predictive modeling is expected to transform our ability to design new waste forms and therefore enable rational discovery strategies for nuclear waste management [36,37,39].

## 2. Incorporation of Waste Elements in Crystalline Ceramic Phases

Crystalline ceramic waste forms are developed with the unique capability of efficiently incorporating certain targeted radioactive waste elements, in contrast to glass waste forms, which incorporate a broad collection of radionuclides but less efficient for problematic radionuclides. Because the distinctive features of their crystal structures are compatible with those targeted radionuclides, they are capable of incorporating a sufficient amount of these waste elements. Such compatibility is largely due to chemical substitutions at crystallographic sites and the flexibility of their crystal structure to accommodate substitutions. For instance, iodide is incorporated into the apatite structure by a substitution in the channel sites, which does not and cannot occur in glass waste forms. It is not surprising that these ceramic waste forms have natural mineral analogues that incorporate waste elements and have endured natural processes for many millions of years [6,11,35]. These crystalline phases are often complex solid solutions with various elements occupying multiple crystallographic sites with variable fractions of occupancies. For instance, apatite-structured materials can have hundreds of end-member compositions as a result of multiple site substitutions [21]: A_5_(XO_4_)_3_Z, where A = Na^+^, K^+^, Cs^+^, Mg^2+^, Ca^2+^, Ba^2+^, Sr^2+^, Cd ^2+^, Pb^2+^, Fe^2+^, Fe^3+^, REE^3+,^ and Ac^4+,3+^ (REE = rare earth elements, Ac = actinides), X = P^5+^, Si^4+^, S^6+^, V^5+^, Cr^5+^, As^5+^, Mn^5+^, Ge^4+^, and Z = OH^−^, F^−^, Cl^−^, Br, I^−^, O^2−^, CO_3_^2−^, and IO_3_^−^. In some cases, as a result of different compositions, the apatite crystal structure can even adapt different symmetries (e.g., $P6_{3},\;P\bar{3}$) through lattice distortion and cation ordering, derived from the hexagonal symmetry (*P6_3_/m*) that natural apatite has [52,53]. Additionally, the usual tetrahedron anion group (XO_4_) can even be replaced by non-tetrahedron anions such as ReO_5_ [54] and BO_3_ [18]. Such structural flexibility and the associated compositional complexity not only allow the apatite structure to incorporate radioactive elements, but also may permit them to tolerate thermodynamic instability caused by a change in the composition resulting from radioactive decay (i.e., β decay). For instance, apatite-structured materials are predicted to be able to mitigate the instability caused by the chemical composition change as a result of β decay from fission products [55]. This mitigation relies on the material’s ability to reduce variable valence metal cations via receiving the electrons emitted from β decay and on the structural accommodation of the changes in the valence and chemical identity of the radionuclides. Overall, these studies suggest that structural flexibility and compositional complexity can greatly benefit the immobilization of radionuclides. 

To fully appreciate the flexibility and complexity of those complex oxides for waste form development, the chemical composition of a given waste form can be further optimized for its performance. Improving the loading capacity of radionuclides and the chemical durability of waste forms is one of the active research areas in nuclear waste form development. Thermodynamic principles, along with computational techniques and calorimetry measurements, have been used to guide ceramic waste form development. Searching for better-performing waste forms typically involves varying chemical substitutions in the host phase [56,57,58,59,60,61,62]. However, the compositional space is often large for complex ceramics. The trial-and-error method using experiments can only be performed for a limited number of compositions. With the help of computational techniques, it is possible to overcome this challenge. For instance, machine learning and first-principles modeling have been employed to screen and evaluate a large number of potential new compositions [21,63,64,65,66,67]. The computational techniques can thus be helpful to complement the experiment, and, in particular, to possibly identify promising candidates from a vast collection of compositional possibilities for further costly experimental investigations. Apatite- and hollandite-structured materials are highlighted here in this contribution as examples and model systems to demonstrate the application of machine learning for nuclear waste form development and apatite-structured materials for the application of first-principles calculations and the modeling of dissolution kinetics. 

## 3. Machine Learning for Ceramic Waste Form Design

### 3.1. Artificial Neural Network Simulation

Machine learning techniques have recently become widely used for material discovery [44,45,47,68,69,70,71,72]. Among them, the artificial neural network (ANN) is a supervised machine learning technique based on statistical principles. The approach is inspired by biological neuron assemblies, their way of encoding and solving problems, and how neurons function in the human brain. The perceptron learning algorithm used in ANN accepts input, performs a computation on the input, and then produces an output [73,74]. As shown in Figure 1, the input data with weight and bias are first passed to the neurons in the hidden layers and processed by a training function, and the data are then passed to the output neuron. The weights and biases are continually adjusted during the ANN simulation to match the predicted results to the actually observed ones [75]. To deduce the relationship between the input and output, a neural network is first trained using a given set of input-output datasets through supervised learning. After the training (supervised learning), the network needs to be validated before it can be used for prediction with an input–output characteristic approximately equal to the relationship of the training problems. Because of the modular and nonlinear activation functions, the network is in principle able to approximate any arbitrary relationship to an arbitrary degree of accuracy [75,76,77]. For the application of ceramic waste forms with a given crystal structure, the composition and composition-derived properties are used as input parameters. The output parameters are selected to have prediction powers such as crystal structure features and thermodynamic properties that can be used to predict potential new compositions. This approach has been used to predict the compositions of iodoapatite and cesium hollandite and the chemical durability of pyrochlore [21,67,78] and is expected, to the first order of approximation, to be able to aid the development of other nuclear waste forms for the prediction of potential new waste form compositions.

### 3.2. Artificial Neural Network Simulation for Ceramic Waste Forms: Cases for Apatite-Structured and Hollandite-Structured Materials

Apatite-structured materials: ANN was used to predict new apatite-structured compositions, including iodoapatite [21,66,79]. To apply ANN to predict new apatite compositions that incorporate iodide, a dataset of the compositions and fully characterized crystal structures of 86 apatite compositions was compiled and used for training and validation [21]. Six parameters, i.e., the average ionic radius and electronegativity of the elements at the A, X, and Z sites of apatite structure A_5_(XO_4_)_3_Z were used as inputs. For a large anion, iodide (I^−^), with a radius of 2.2 Å, to be incorporated in apatite structure, the channel site of the apatite needs to have the right size so that there is no mismatch between iodide and the channel. It was hypothesized that apatite compositions whose channels accommodate the iodide ions (as a spherical particle) without mismatch are the most likely chemical compositions for incorporating iodide. This is a reasonable assumption due to the highly ionic nature of iodide bonding in the channel site. Therefore, channel size is likely a good indicator of possible iodine incorporation and was used as the output of the neural network. As shown in Figure 2, the channel size was predicted from the trained network. Using 3.5% as the flexibility of the structure to accommodate iodide, the compositions for which the channel sizes are located between the purple lines are predicted to be potential apatite-structured materials incorporating iodide. The result suggests that combinations of A-site cations of Ag^+^, K^+^, Sr^2+^, Pb^2+^, Ba^2+^, and Cs^+^, and X site cations of Mn^5+^, As^5+^, Cr^5+^, V^5+^, Mo^5+^, Si^4+^, Ge^4+^, and Re^7+^, are possible apatite compositions that can incorporate iodide at the Z site. This prediction is consistent with existing data from the literature based on experimental synthesis and first-principles calculations. As shown, iodoapatite Pb_5_(VO_4_)_3_I has been synthesized [16] and the predicted channel size is within the values of possible iodoapatite compositions and is ~0.1% from the experimentally determined channel size (Figure 2). Additionally, iodoapatite Ba_5_(VO4)_3_I and Sr_5_(As_4_)_3_I were predicted to be potential apatite compositions, which is in agreement with first-principles calculations (discussed in Section 4.1). Recent experiments have also confirmed some of these predictions [80,81]. For instance, arsenate–lead iodoapatites Pb_5_(AsO_4_)_3_X (X = OH, Cl, Br, I) have been synthesized from the precipitates of solution at acidic pH and ambient temperature conditions. Their structure and compositions were well characterized and thermodynamic properties such as enthalpy were measured using melt drop solution calorimetry [80]. In addition, iodoapatite (Ba_5_(VO_4_)_3_I) has been synthesized using a high-energy ball milling machine and spark plasma sintering technique [81].

Hollandite-structured materials: A similar strategy and method were used to predict compositions of hollandite for the incorporation of cesium-137 [67]. Chemical substitutions in hollandite (A_2_B_8_O_16_) can occur at both A and B sites, where the A site can be occupied by alkali and alkaline earth elements such as Na^+^, Cs^+^, Rb^+^, and Ba^2+^, and the B site can be occupied by various di-, tri-, and tetravalent cations such as Mg^2+^, Fe^2+^, Fe^3+^, Al^3+^, Cr^3+^, Ti^3+^, Ti^4+^, and Si^4+^. Both sites can have substantial substitutions to form solid solutions. The A site in the tunnel can accommodate large cations such as cesium and barium. An interesting characteristic of the hollandite structure is that its tunnel size is largely controlled by the B site cations. A site is often partially occupied with vacancies, but both the O and B sites are usually fully occupied. The normal charge is balanced by coupled substitutions at both the A and B sites. For ANN simulations, only four parameters, i.e., the average ionic radius and electronegativity of the A and B sites of hollandite, A_2_B_8_O_16_, were used as inputs (because the oxygen sites always remain fully occupied, the inputs for oxygen sites do not vary with A and B site compositional variance). The number of vacancies was not used as an input and including it did not improve the prediction. Since the objective is to find the appropriate channel size that can accommodate large cations such as Cs, the channel size is used as the output for the structural property, similar to the channel size used for iodide incorporation in iodoapatite [21]. A structural stability criterion, called the tolerance factor, was also used to further narrow the ANN-predicted compositions in addition to the channel size. The tolerance factor criterion is often used in experimental studies to rule out candidate structures that are potentially less plausible [60,82]. Figure 3 shows ANN-predicted possible compositions defined by the tolerance factor and channel size of hollandite-structured materials. The data encompass a combination of Ba and Cs with varying A-site occupancy and B-site compositions. Given the tolerance factor between 0.90 and 1.10 and channel size between 2.80 and 3.15 Å, which are reasonable ranges based on the experimental observations in our dataset, the compositions located within the rectangular box (Figure 3) can be considered possible hollandite compositions. A wide range of previously unexplored Cs–hollandite compositions were evaluated as possible, including M^4+^ = Zr^4+^ and Sn^4+^ at the B site. These compositions are likely to be potential candidates for immobilizing Cs based on the ANN predictions of their channel size [67]. In particular, a combination of some of the aforementioned variable-valence M^3+,2+^ and M^4+,3+^ cations, such as Fe^3+,2+^ and Ti^4+,3+^, has also been predicted to be highly possible. These hollandite compositions can also be candidates for accommodating the chemical changes as a result of the β decay of Cs-137 due to radioparagenesis [55,83]. Although Ti-based hollandite compositions are the most extensively investigated, encompassing most recent studies [56,57,58,59,60,61,84], several new Ti-based hollandite compositions were also predicted to have potential for the immobilization of Cs-137 [67].

As summarized above, the progress made on apatite and hollandite compositions suggests that the compositional space of these phases remains highly underexplored and could be a fruitful area of research in ceramic nuclear waste development. In addition, there are many different complex oxides with structural and compositional complexities similar to apatite-structured and hollandite-structured materials, such as murataite, zirconolite, perovskite, garnet, crichtonite, zeolite, pollucite, sodalite, and monazite, to name a few. Due to the possible compositional variance of different crystallographic sites by chemical substitution to form solid solutions, the number of possible unique compositions is very high, and the compositional space is vast. It is thus practically impossible to experimentally investigate the entire compositional space for any of those crystal structures. Instead, it requires synergistic effort from both the computational and experimental methods in order to accelerate ceramic nuclear waste form development. For instance, from the computational front, as demonstrated for apatite and hollandite phases, the ANN studies have predicted several unexplored potential compositions that could be further explored by experiments.

## 4. First-Principles Thermodynamics and Electronic Structure Calculations for Ceramic Waste form Development

### 4.1. First-Principles Thermodynamics of Ceramic Waste Forms and the Case for Apatite-Structured Materials 

The predictions from the artificial neural network are based on a supervised machine learning approach, and the results essentially provide a collection of potential compositions that may exhibit favorable characteristics when incorporating certain radionuclides. Although artificial neural network simulations are able to match compositions to properties, they do not provide the physical reasons or mechanistic understanding for the prediction and thermodynamic stability of the predicted compositions. In order to estimate the stability, first-principles calculations can be employed to compute thermodynamic properties with respect to alternative phases. Such computational methods have been used for ceramic nuclear waste from development, such as nuclear waste forms for the radionuclide Sr-90 [63,85]. In this method, the free energy or enthalpy of formation of the phases is calculated from the crystal structure and composition using first-principles methods. Potential energy is calculated from the energy optimization of the structure at zero kelvin. The zero-point energy and vibrational entropic contribution to the total free energy at finite temperature are then calculated from the phonon density of states and quasi-harmonic approximation using the following equation:(1)$$F(T,V)=E(V)+\int_{0}^{\infty }d\omega g(\omega )\left(\frac{1}{2}\hbar \omega +K_{B}Tln(1-e^{-\frac{\hbar \omega }{K_{B}T}})\right)$$
where *F(T,V)* is free energy, *E(V)* is the potential energy calculated at zero kelvin, *K_B_* is the Boltzmann constant, *T* is temperature, and *g(ω)* is the phonon density of state. This method is well-established and has been applied in materials science in various applications [63,86,87] and for ceramic nuclear waste development [85,88]. Here, an application of the method for iodide in apatite-structured materials is highlighted.

Apatite: To illustrate this methodology for iodoapatite application, compositions of Sr_5_(AsO_4_)_3_I and Ba_5_(VO_4_)_3_I were selected, which are based on their potential candidacy as predicted by artificial neural network simulations [21], but previously have not been synthesized experimentally. A typical first-principles calculation for thermodynamic stability is based on density functional theory with plane-wave basis sets using GGA PBE exchange-correlation. For this example, the program CASTEP was used with a Monkhorst–Pack grid for k-point sampling and ~1 k-point per ~0.1 Å^−1^ in each dimension. The cutoff energy of 650 eV was used in the calculations. The convergence criteria for the force were set to 0.01 eV/Å for the geometry optimization and 0.001 eV/Å for phonon calculations. For Sr_5_(AsO_4_)_3_I iodoapatite, similar to Pb_5_(VO_4_)_3_I, two possible synthesis routes are possible: from oxides and a simple salt (Equation (2)) or from an intermediate phase Sr_3_(AsO_4_)_2_ and a simple salt (Equation (3)): 9 SrO + 3 As_2_O_5_ + SrI_2_ = Sr_10_(AsO_4_)_6_I_2_(2)
3 Sr_3_(AsO_4_)_2_ + SrI_2_ = Sr_10_(AsO_4_)_6_I_2_(3)

The iodoapatite can also decompose (i.e., the reverse reactions of Equations (2) and (3)). The calculated free energy of reactions is plotted in Figure 4 for each of the reactions. The result suggests that Sr_10_(AsO_4_)_6_I_2_ iodoapatite is thermodynamically stable with respect to both the oxides (or the simple salt) and the intermediate (Sr_3_(AsO_4_)_2_) at temperatures below ~410 K.

This result is consistent with experimental observations on the thermal stability analysis of iodoapatite samples synthesized using different methods. Pb_5_(VO_4_)_3_I decomposes to vanadate (Pb_3_(VO_4_)_2_) at a temperature of ~540 K for a sample synthesized using a hot press [19]. The thermal stability is enhanced to ~940 K for a dense ceramic synthesized by spark plasma sintering [64]. While the free energy for the reaction based on Equation (2) decreases with temperature (Figure 4a), the free energy for the reaction based on Equation (3) increases with temperature (Figure 4b). The results suggest that there is a subtle temperature effect on the thermodynamic stability of the iodoapatite and the intermediate phase. An appropriate temperature range based on these calculations can be realized as a guide for the design of the experimental synthesis. 

### 4.2. Electronic Structure Calculations of Ceramic Waste Forms and Cases for β-Decay-Induced Instability (Radioparagenesis) in Apatite-Structured Materials

β decay of fission products such as Cs-137 and Sr-90 in ceramic waste forms presents a great challenge for waste form design. As these elements decay, a change in chemical identity occurs, which can cause thermodynamic instability of the materials [89,90,91,92]. A hypothesis was proposed, which states that the structural and energetic instability caused by β decay can be mitigated by introducing variable valence cations as the electron acceptor in the host material at neighboring crystallographic sites with a flexible crystal lattice, as shown in Figure 5 [55]. As a demonstration, apatite-structured materials were considered for Cs-137 and Sr-90 incorporations to test this hypothesis. Ferric iron was used as an electron acceptor. DFT calculations were performed to calculate the electron density of states, local defect structure, and energetics. The localization of the β electron captured by the ferric ion is robust for all the compositions investigated. The magnitude of the structural distortions at the sites occupied by iron and transmuted elements (Ba-137, Y-90, and Zr-90) is small to moderate. The minor structural changes and favorable energetics indicate the stability of the materials after beta transmutations. The results suggest apatite-structured materials could be promising nuclear waste forms to mitigate the β-decay-induced instability by incorporating variable valence cations such as ferric iron into the structure. This study provides a new insight into the development of nuclear waste forms: it is possible to incorporate fission products undergoing β decay in a crystalline phase. This methodology and the strategy for the localization of the β decay effect could be applied to other nuclear waste forms for incorporating β-emitting fission products. These theoretical results can be experimentally tested using short-lived radionuclides undergoing β decay.

As stated earlier, first-principles calculations are a well-established tool in materials science and engineering. They have been essential in the discovery of new materials for various applications, which is highlighted in the Materials Genome Initiative [93]. The application of this methodology to nuclear waste form development is expected to accelerate the optimization of existing nuclear waste forms and the discovery of new waste forms with improved properties. This can be achieved by guiding the materials’ synthesis based on machine learning and calculated thermodynamic properties of the materials using high-performance computers, as demonstrated by these examples.

## 5. Modeling of the Dissolution Kinetics of Crystalline Ceramics in Aqueous Solution

### 5.1. Kinetic Rate Theory for Dissolution of Minerals and Crystalline Ceramics

The prediction of the dissolution rate of nuclear waste forms in the environment is essential for nuclear waste management. Decades of research have resulted in the development of a collection of ceramic waste forms for various radionuclides. However, characterizing their chemical durability and predicting their long-term dissolution rate still present a great challenge to the scientific community. First of all, the dissolution rate is not an intrinsic property of a material. Rather, it is a response of the intrinsic properties associated with dissolution kinetics to the perturbations of the environmental variables through the associated surface reactions [39]. Thus, determining these intrinsic properties is essential for the characterization of the chemical durability and the prediction of the dissolution rate in the environment. 

Thermodynamic rate equations provide the fundamental basic law for modeling and predicting dissolution kinetics [94,95,96,97]. In these equations, environmental conditions are described by the corresponding variables, and the properties that are intrinsic to the specific material in consideration are constants in the following rate equations:(4)$$r=k_{+}\cdot \left(1-e^{\left({∆G}_{r}/RT\right)}\right)$$
(5)$$k_{+}=k_{0}\cdot {\left[H^{+}\right]}^{\eta }\cdot \prod_{i}{a_{i}}^{v_{i}}{\cdot e}^{-E_{a}/RT}$$
where R is the gas constant. $T,\;\left[H^{+}\right],\;{and\;a}_{i}$ are temperature, *H^+^* activity, and the activity of aqueous species i, respectively, which are environmental variables considered in the rate equation. $k_{+},{\;∆G}_{r},\;k_{0},\;\eta ,\;v_{i},{and\;E}_{a}$ are the forward rate, the Gibbs free energy of the dissolution reaction, the rate constant, the reaction order involving *H^+^*, the reaction order involving species i, and the activation energy, respectively. These factors control the intrinsic properties of the material for dissolution. If these constants are determined by either theory, simulation, or experiment, the chemical durability can be characterized, and the dissolution rate can then be predicted using the rate equations. However, these constants are not all known for many ceramic waste forms. In the literature, different leach test protocols have been employed to study the dissolution of nuclear waste forms, making it difficult to compare the reported results and to use them in the rate equations for prediction. To characterize the chemical durability of nuclear waste forms, standard test methods such as the Product Consistency Test [98] have been used to estimate the chemical durability of various glass and ceramic waste forms. However, these standard methods are often designed to extract elemental release rates and are used for screening candidate waste forms under given conditions rather than determining the intrinsic properties that define the thermodynamic chemical durability of the materials as defined in the rate equations (Equations (4) and (5)). Several experiments are necessary to provide all the intrinsic properties relevant to dissolution kinetics, which are required to predict the dissolution rate under various environmental conditions. 

### 5.2. Application of Kinetics Rate Theory—A Case Study of Iodoapatite Waste Form

Progress has been made in the understanding of iodoapatite dissolution [99,100,101], complex interactions at microscopic scales [102,103], interactions with other materials [104,105], and the prediction of the dissolution kinetics of iodoapatite using rate equations [1,106]. These experiments suggest the importance of complex interactions and processes involving iodoapatite dissolution such as ion exchange, diffusion, and transport through grain boundaries, in addition to dissolution. By including a time-dependent diffusion-controlled process and a parameter related to the solution saturation, the equations (Equations (4) and (5)) were modified to
(6)$$r\left(t,\;p_{s},T,pH\right)=r(t)\cdot k_{0}\;{\cdot 10}^{-\eta \cdot pH}\cdot [e^{-E_{a}/RT}]\cdot [1-e^{-\phi p_{s}}]$$
where $t,\;p_{s},\;T,\;and\;pH$ are time, the ratio of the reacting solution volume to surface area (volume-to-surface-area ratio), solution temperature, and solution *pH* respectively. The constants $k_{0}$, *η*, *R*, *E_a_*, and *ϕ* are the rate constant, the order of dissolution reaction with respect to *pH*, the gas constant, the activation energy, and a constant related to saturation, respectively. The r(t) term in Equation (6) describes the time dependence of iodine diffusive release [1,106,107,108]. The last term in Equation (6) replaces the second term in Equation (4), which was based on the simple assumption that the Gibbs free energy of dissolution is approximately proportional to the solution volume-to-surface-area ratio. A series of experiments were systematically designed to determine the constants (i.e., the intrinsic properties that define the chemical durability) in the rate equation (Equation (6)). Once these constants are determined, the equation (Equation (6)) can be used for predictions [1]. As illustrated in Figure 6, iodine release rates are predicted at various environmental and test conditions, including time, pH, temperature, and volume-to-surface-area ratio. One of the significant findings is that the iodine release rate is bounded by diffusion-controlled kinetics at the lower end of the dissolution and by far-from-equilibrium dissolution kinetics at the higher end [1]. In addition, the long-term iodine release rate is significantly lower than the rate measured in short-term laboratory tests. The short-term rate is largely controlled by a transient diffusion process and the surface effect. The result demonstrates that it is possible to consider all critical processes that determine dissolution kinetics and to parameterize the rate equations with the variables relevant to the environmental conditions. The parameterized rate equations can then be used to predict the performance of a ceramic waste form under various environmental conditions.

The above example demonstrates a strategy in which each of these intrinsic properties (i.e., the constants in the rate equations) are estimated from a series of critical experiments that are systematically designed for the environmental variables. However, such a method still needs a substantial number of experiments for the strategy to work. It is desirable that these intrinsic properties in the rate equations be predicted from the composition and structure of materials alone. Although great effort has been dedicated to estimating some of these properties, only a few examples have been reported to predict some of them, such as the forward dissolution rate of isostructural families of divalent metal oxides and orthosilicates [109] and the reaction order of the pH dependence of silicates, aluminosilicates, and quartz [110]. It is imperative that these intrinsic parameters within a group of materials and across different groups be predicted using the properties defined by the structure and composition only without costly experiments in order to accelerate quantitative predictions of the dissolution kinetics of materials. 

## 6. Computational Materials Design and Performance Prediction of Ceramic Nuclear Waste Forms

Understanding the mechanisms of the incorporation of the waste elements in ceramic waste forms and their dissolution kinetics in the environment plays an important role in nuclear waste management. Such an understanding is essential for the performance evaluation of nuclear waste forms in the environment. Examples of ceramic waste forms based on apatite-structured and hollandite-structured materials demonstrate a methodology with which new iodoapatite and Cs–hollandite compositions were predicted using a combination of artificial neural network simulations and first-principles calculations. The dissolution rates were predicted under various environmental conditions using thermodynamic rate equations. By integrating the predictions of the compositions of waste forms and their performance in aqueous solutions in a holistic framework, this strategy could be used to design ceramic waste forms with optimal incorporation capacity and environmental performance by varying the chemical compositions.

Figure 7 summarizes this approach, which can provide a path for accelerated ceramic waste form development and performance prediction for ceramic nuclear waste forms. In this approach, material design and performance modeling are considered simultaneously. The design of ceramic waste forms considers the loading capacity of waste elements and focuses on problematic radionuclides. Certain crystalline ceramics can be targeted using thermodynamic principles in combination with machine learning and first-principles calculations. Performance modeling considers the dissolution kinetics associated with rate-determining critical processes using rate equations. The intrinsic properties in the rate equations are parametrized using a series of experiments or predicted based on relationships between structure, composition, and properties. Critical experimental tests are carried out to understand the dissolution reactions and benchmark intrinsic properties predicted by theories. The verification of modeling results outside the range of available data can be performed to validate the predictions of the modeling. Models and model parameters are open for revision as new data emerge. The verified predictive models will be able to predict new waste forms, their incorporation capacity, and long-term dissolution performance. The composition of a given waste form can then be optimized in terms of immobilization efficiency and chemical durability.

The methodology outlined above for apatite-structured and hollandite-structured materials is also suitable for other complex materials with chemical substitutions on multiple structural sites. Materials such as perovskite, pyrochlore, murataite, monazite, and crichtonite, although dissimilar to apatite-structured and hollandite-structured materials in structure and composition, also have multiple crystallographic site substitutions, flexible crystal structures, and complex chemical compositions. These materials are suitable for the immobilization of various waste streams and waste elements, including fission products, actinides, and activation products. While the structural flexibility and compositional complexity offer plenty of room for the optimization of these nuclear waste forms, the composition space becomes too populated to be handled by experiments alone. It is hoped that the approach proposed here of combining computational approaches with predictive modeling will provide a robust strategy to accelerate the development of ceramic waste forms.

## 7. Summary and Conclusions

Ceramic waste forms are developed to immobilize radionuclides by considering the effectiveness of the incorporation of radionuclides, mechanical and chemical stability in the environment for a long period of time, and cost and technical readiness in practical application. Due to their unique crystallographic features and behavior in nature environment as demonstrated by their natural analogues, ceramic waste forms are suitable to target a group of or a single radionuclide due to their compatibility with the host structure. Recent progress is highlighted in this review, focusing on using computational and modeling approaches to accelerate the development of ceramic waste forms. To overcome challenges of the enormous compositional space in complex ceramic waste forms because of multiple substitutions at their crystallographic sites, machine learning is employed to narrow the scope of potential compositions and to search for new compositions that are consistent with their crystallochemistry. Such an approach has been applied to apatite, hollandite, and pyrochlore. Some predictions were recently verified in the experimental synthesis of arsenate–lead iodoapatite and barium–vanadium iodoapatite compositions. In addition, first-principles thermodynamics computations are employed to calculate the thermodynamics stability of the compositions predicted from machine learning, a further reduction in the compositional space for a given waste form. Calculations on an iodoapatite form illustrate a computational approach that predicts the stability of Sr_10_(AsO_4_)_6_I_2_ apatite with respect to its intermediate phase Sr_3_(AsO_4_)_2_. Electronic structure calculations of apatite-structured materials demonstrate that the structural and energetic instability caused by β decay can be mitigated by introducing variable valence cations as the electron acceptor at neighboring crystallographic sites in a complex host phase with a flexible crystal lattice such as apatite structure. The chemical durability and dissolution kinetics are modeled using kinetic rate theory with parameters extracted from critical experiments, as demonstrated for a Pb_10_(VO_4_)_6_I_2_ apatite sample. By integrating the predictions of elemental incorporation and thermodynamic stability and the modeling of dissolution kinetics in a holistic framework, this approach as highlighted in this review could be promising to accelerate ceramic waste form development based on complex crystalline phases, as well as the performance prediction for problematic nuclear waste elements.

## Figures and Tables

**Figure 1 materials-16-04985-f001:**
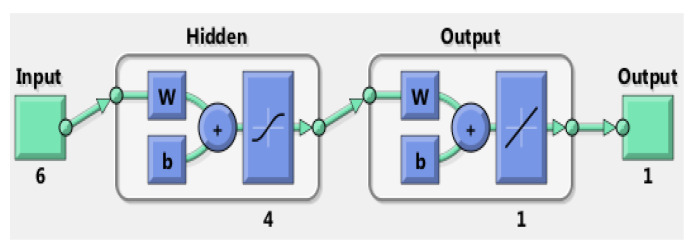
A schematic diagram of an artificial neural network. W and b are weights and biases for activation function and output. In this diagram, there are 6 input parameters and 1 output parameter, 1 output layer, and one hidden layer with 4 neurons used in the hidden layer. Adapted from [21].

**Figure 2 materials-16-04985-f002:**
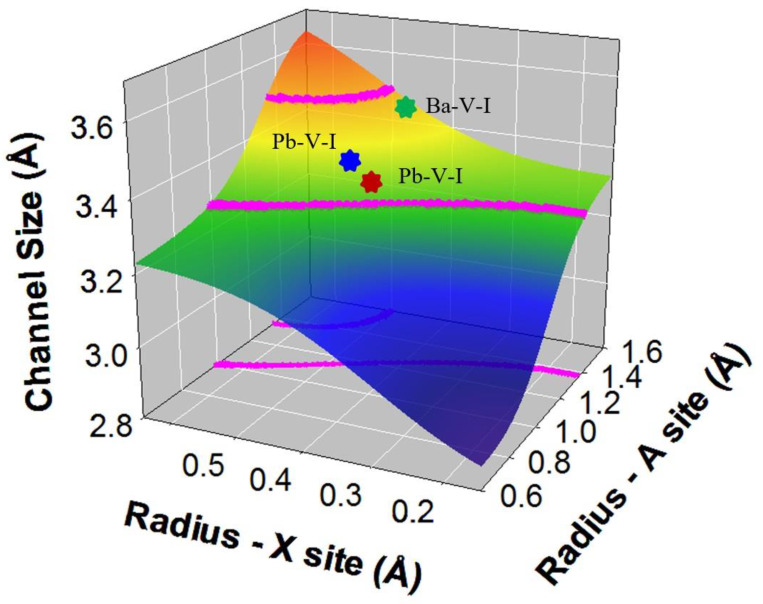
Prediction of iodoapatite compositions with iodide ions in the structural channel site. The region between the pink lines defines the average radii of possible A and X cation combinations. The channel size is calculated based on the ionic radius of iodide (2.20 Å) and coordination cations and a 3.5% prediction error. The region is also both projected onto the radius-A and radius-X planes and onto the surface of predicted channel size. The stars indicate the locations of Pb_5_(VO_4_)_3_I (blue), Ba_5_(VO_4_)_3_I (green), and Sr_5_(As_4_)_3_I (red) apatite compositions. Modified and adapted from [21].

**Figure 3 materials-16-04985-f003:**
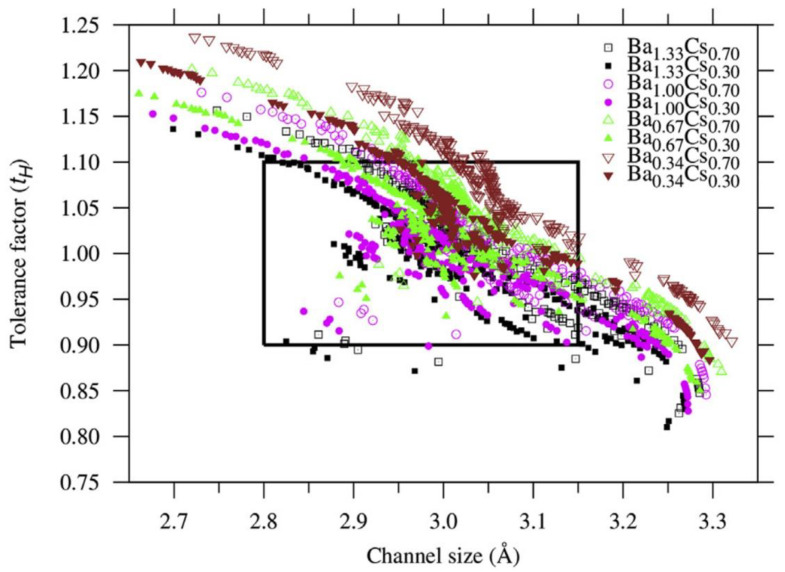
Prediction of Cs–hollandite compositions based on the channel size and the tolerance factor (τ_H_) with a combination of Cs and Ba at the A site. B-site compositions comprise combinations of 3^+^ and 4^+^ cations from the entire dataset considered in this study. Adapted from [67].

**Figure 4 materials-16-04985-f004:**
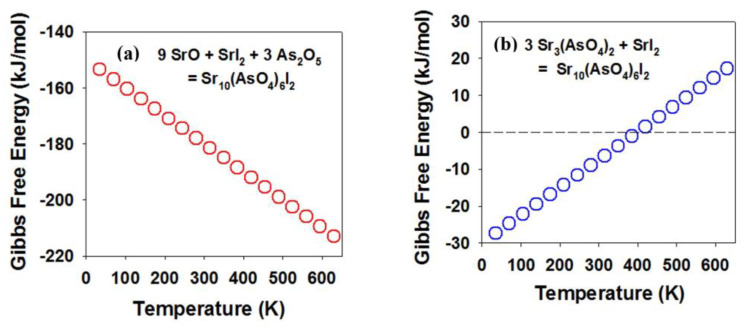
Formation free energy of iodoapatite Sr_5_(As_4_)_3_I from the oxides and simple salts (**a**) and from the intermediate Sr_3_(AsO_4_)_2_ (**b**) as a function of temperature based on first-principles calculations. The red (**a**) and blue (**b**) circles are calculated Gibbs Free Energy of reaction (Equation (2)) and reaction (Equation (3)) respectively.

**Figure 5 materials-16-04985-f005:**
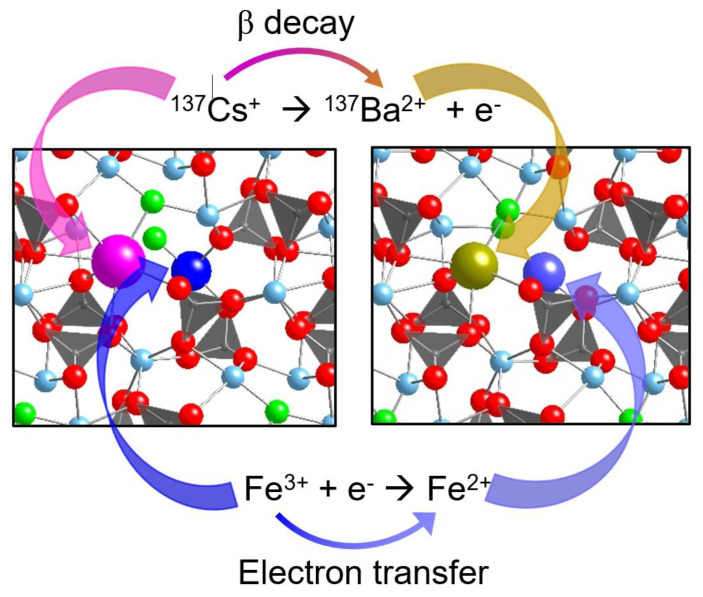
The schematic of the electron transfer from the β decay of Cs-137 to the neighboring ferric ion in apatite. The ferric ion is changed to a ferrous ion. Cs^+^ (large purple ball), Ba^2+^ (large yellow ball), Fe^2+^ (large blue-purple ball), and Fe^3+^ (large blue ball) were modeled as the substituted impurity of Ca in Ca_5_(PO_4_)_3_F apatite. Dark-grey polyhedrons are [PO_4_], small red balls are O, small light blue balls are Ca, and green balls are F. Modified and adapted from [55].

**Figure 6 materials-16-04985-f006:**
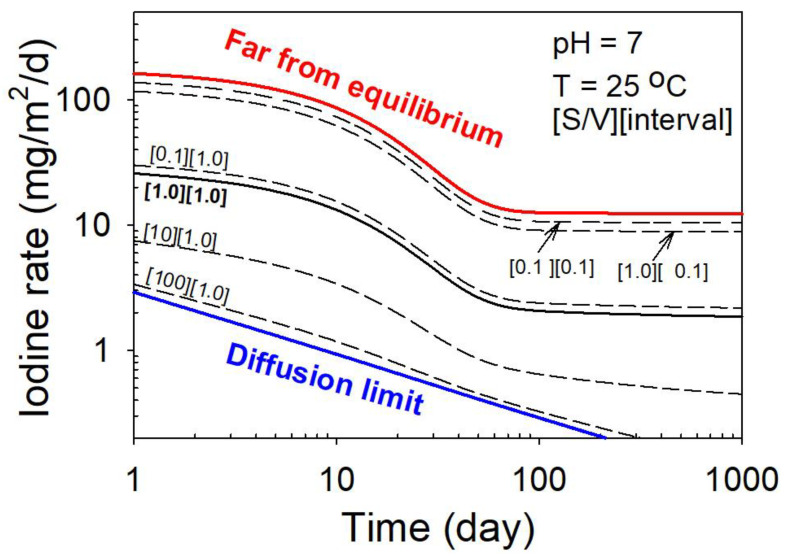
A prediction of iodine release rate as a function of time at different S/V ratios (m^−1^) and intervals (days) under a pH of 7 and temperature of 25 °C. The S/V ratio and interval are indicated. The lines are predictions. The high release rate line marks the far-from-equilibrium limit (thick red line), and the low release rate line is the diffusion limit (thick blue line). Modified and adapted from [1].

**Figure 7 materials-16-04985-f007:**
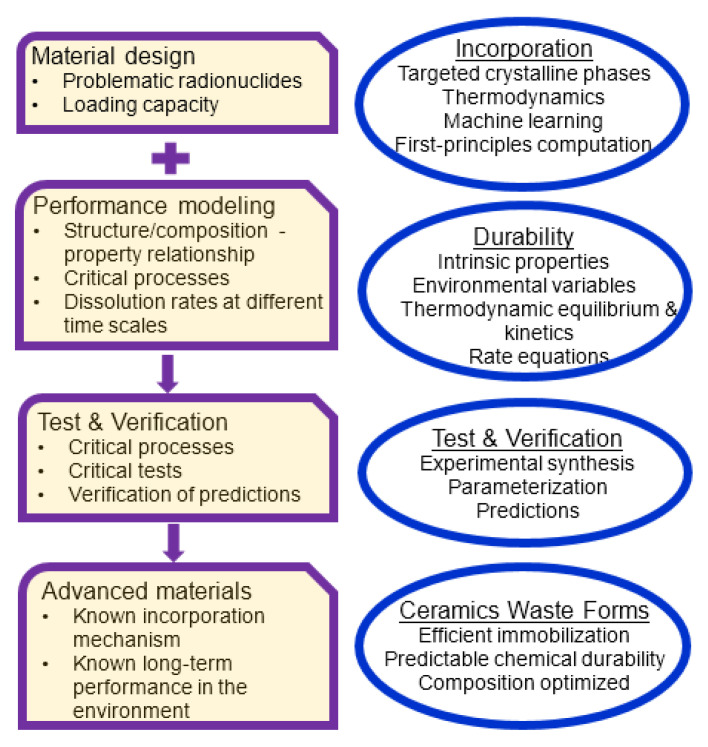
A schematic diagram for the development of advanced ceramic waste forms based on a holistic approach by simultaneously considering material design and performance prediction and model verification.

## Data Availability

The raw data required to reproduce these findings are available upon request from the authors.

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
