# Peer review of "Computational Materials Design for Ceramic Nuclear Waste Forms Using Machine Learning, First-Principles Calculations, and Kinetics Rate Theory"

_materials, 2023, doi:10.3390/ma16144985_

Round 1

Reviewer 1 Report

Review for the article “Computational materials design for ceramic nuclear waste forms using machine learning, first-principles calculations, and kinetics rate theory”

The article is dedicated to a long-known problem of safe nuclear waste disposal. The question of radioactive materials burial had recently got a renewed attention due to a global interest in both ecology and energy solutions.

The nuclear fuel cycle starts with a mining, purification and enrichment procedures leading to a constant and manageable compositions, on the next step the nuclear fission is used for the energy generation. This step produces a highly radio- and chemically active material with the composition not only unpredictable, but also changing in time. This kind of substance has an immense toxicity and is a threat for human and animal health and ecosphere for hundreds of years.

Thus, the nuclear waste requires an assured containment which should respond a number of criteria: chemical stability, ability to incapsulate various ions inside its structure and certain stability to a metamictization process and large-scale production and low price.

The immense toxicity of nuclear waste requires a guaranteed failure-proof solutions, which in turn attracts scientists to search for the solution in Geology, and more precisely in Mineralogy of REE, U and Th.

The first question arises in Introduction, where authors state that “Surfaces of many silicate minerals, common in the disposing environment, are negatively charged” which needs a little description, namely in a few words should be mentioned acidic and basic rocks, both of which are mostly composed of silicates and justify a reference to natural materials though no natural materials are discussed later.

The second group of questions relates to the selected group of materials. Despite the fact that the review is aimed at applying machine learning methods to ceramic materials, the article is devoted to the study of iodoapatite and, to a much lesser extent, hollandite. In this regard, there are questions about the title of the article, the rationale for the choice of a particular materials. In addition, the article mentions the name “apatite” many times, but only in relation to a group of synthetic materials that have the only common element with original apatite - oxygen. Please consider clarifications such as “apatite-structured” or “apatite-based”

Also, in the list of references, less than a quarter of the sources were published in the last 5 years, which is normal for a review article on a narrow topic if the main works covering recent achievements are covered. However, quite significant works can be found on the topic (for example, https://www.nature.com/articles/s41529-020-0119-9, https://doi.org/10.1016/j.commatsci.2021.110820), adding information on ion exchange in the selected material in addition to the dissolution described in the article and supplementing information on ceramic materials for nuclear waste.

And in the conclusions section after the article devoted to apatite-like materials, the authors expand data imports to structurally dissimilar materials - perovskite, monazite, crichtonite, etc., which have radically different stability, chemical behavior, and structural types.

Reviewer 2 Report

The work is interesting and keeps pace with the times. Using a computer to solve technological problems sometimes makes it possible to significantly reduce the number of experimental works.

Recommendation for the Introduction section:

Ceramic forms of radioactive waste is a promising topic, which, however, has a number of significant problems when considering the possibility of practical implementation. First of all, it is impossible to talk about waste ceramization without really working fractionation, because the waste has a complex chemical and radionuclide composition, a certain ratio of components. Ceramics is not omnivorous to the immobilization of waste components (unlike glass melt), the possibilities of including various mixtures of radionuclides are limited. There are technological issues: the lack of real industrial equipment, the technology for placing forms in protective containers is unclear, the containers themselves have not been developed, and many others. In addition, practically nothing has been said about long-lived and highly toxic isotopes of actinides. At the same time, ceramics is especially promising specifically for actinide wastes, and the problem of fission products (first of all, relatively short-lived cesium and strontium isotopes) can be dealt with using other matrices.

I think that it is necessary to give such an addition so that it is clear that the authors are talking more about work for the future, and not about solving urgent problems of the nuclear fuel cycle.

Notes:

- about 30% of the sources in the bibliography are 15 years and older; it is recommended to evaluate the need for references to older works$

- Check text font size

Reviewer 3 Report

This paper describes the computational materials design for ceramic nuclear waste forms using machine learning, first-principles calculations, and kinetics rate theory. The methodological approach seems effective and appropriate. From this work, it provides an accelerated path for ceramic waste form development and the performance prediction for problematic nuclear waste elements. The paper was well organized, and the results are important for material science. Therefore, I feel this paper should be acceptable after some revisions in view of the following specific comments.

(1)   It would be interesting if some examples of experimental verification can be presented alongside computational results.

(2)   It would be better for the authors to give a summary of this manuscript in appropriate section, that is, please make “Conclusion” Section at the end of the manuscript.

There is basically no problem regarding English.

Reviewer 4 Report

The review manuscript titled "Computational materials design for ceramic nuclear waste forms using machine learning, first-principles calculations, and kinetics rate theory" is an exemplary piece of scientific writing, showcasing meticulous attention to detail and a comprehensive description of each section. The authors have demonstrated a high level of rigor and precision throughout their work. Of particular note is the authors' adeptness in the computational aspects, as they have provided a multitude of critical comments and insightful analyses. These comments not only enhance the overall quality of the paper but also highlight the authors' commitment to scientific integrity and robustness. Their proficiency in addressing computational challenges further strengthens the validity of their findings.

Few minor corrections should be made to the text formatting, especially in the Introduction part. I suggest the Authors to carefully check all sections for minor text formatting issues.

Therefore, I recommend the publication of the manuscript on Materials

Round 2

Reviewer 3 Report

Based on the reviewers' comments, the manuscript is properly corrected on the whole. The study will be prospective and contributes for the field of energy and ecology in future studies. Therefore, I feel this paper should be acceptable for publication.

There is basically no problem regarding English.